# Hematological Changes in Gas Station Workers

**DOI:** 10.3390/ijerph20105896

**Published:** 2023-05-20

**Authors:** Isabela Giardini, Katia Soares da Poça, Paula Vieira Baptista da Silva, Valnice Jane Caetano Andrade Silva, Deborah Santos Cintra, Karen Friedrich, Barbara Rodrigues Geraldino, Ubirani Barros Otero, Marcia Sarpa

**Affiliations:** 1Technical Area of Environment, Work and Cancer, National Cancer Institute-INCA, Rua Marquês do Pombal, 125/5º andar-Centro, Rio de Janeiro CEP 20230-240, RJ, Brazil; 2Laboratory of Environmental Mutagenesis, Department of Biochemistry, Biomedical Institute, Federal University of the State of Rio de Janeiro (UNIRIO)-Rua Frei Caneca, 94/4º andar-Centro, Rio de Janeiro CEP 20211-010, RJ, Brazil; 3Centro de Estudos de Saúde do Trabalhador e Ecologia Humana (CESTEH), Escola Nacional de Saúde Pública, Fundação Oswaldo Cruz. Rua Leopoldo Bulhões, 1480-Manguinhos, Rio de Janeiro CEP 21041-210, RJ, Brazil; 4Department of Collective Health Biochemistry, Biomedical Institute, Federal University of the State of Rio de Janeiro (UNIRIO)-Rua Frei Caneca, 94-Centro, Rio de Janeiro CEP 20211-010, RJ, Brazil

**Keywords:** occupational exposure, gas stations, filling station attendants, benzene, hematology, clinical laboratory techniques

## Abstract

(1) Background: Benzene, toluene, and xylene isomers (BTX) are present in gasoline. Exposure to benzene may lead to the appearance of a series of signs, symptoms, and complications, which are characterized by benzene poisoning, which is an occupational disease. This study evaluated the presence of signs and symptoms related to occupational exposure and whether occupational exposure to BTX is associated with the development of hematological changes. (2) Material and Methods: This cross-sectional epidemiological study included 542 participants, in which 324 were gas station workers (GSWs) and 218 were office workers (OWs) with no occupational exposure to benzene. To characterize the type of exposure (exposed and not exposed), trans,trans-Muconic acid (tt-MA), Hippuric acid (HA), and Methylhippuric acid (MHA) were used as exposure biomarkers. The tt-MA analysis revealed that the GSWs had 0.29 mg/g of urinary creatinine and the OWs had 0.13 mg/g of urinary creatinine. For HA, the GSWs presented 0.49 g/g of creatinine while the OWs presented 0.07. MHA analysis revealed that the GSWs had 1.57 g/g creatinine and the OWs had 0.01 g/g creatinine. Occupation habits and clinical symptoms were collected by questionnaire and blood samples were analyzed for hematological parameters. The persistence of hematological changes was evaluated with three serial blood collections every 15 days followed by laboratory hematological analysis. A descriptive analysis by the Chi-square test method was performed to evaluate the association between occupational exposure to fuels and the occurrence of changes in hematological parameters. (3) Results: In the GSWs, the most described signs and symptoms were somnolence (45.1%), headache (38.3%), dizziness (27.5%), tingling (25.4%), and involuntary movement (25%). Twenty GSWs that presented hematological alterations performed serial collections fifteen days apart. In addition, these workers presented total leukocyte counts above the upper limit and lymphocyte counts close to the lower limit. Leukocytosis and lymphopenia are hematological alterations present in chronic benzene poisoning. (4) Conclusions: The results found an initial change in different hematological parameters routinely used in clinics to evaluate health conditions. These findings reveal the importance of valuing clinical changes, even in the absence of disease, during the health monitoring of gas station workers and other groups that share the same space.

## 1. Introduction

Gasoline produced from oil refining is widely marketed at gas stations throughout Brazil [1]. Benzene is a liquid volatile and flammable aromatic hydrocarbon naturally present in crude oil and gasoline [2]. It may also be present in the air due to incomplete combustion of coal and crude oil in the vapors of the steel and petrochemical industries, as well as cigarette smoke [3]. Although environmental exposure to benzene vapors may occur through industrial emissions, exhaust gasses from gasoline-fueled automobiles, coal combustion, and cigarette smoke, occupational exposure is the greatest risk [4], due to frequent contact with benzene during the workday.

In addition to benzene, gasoline contains other volatile organic compounds (VOCs), including toluene and xylene isomers, represented by the term BTX. Workers involved in the production, transport, and commercialization of fuels are exposed to various levels of BTX [2,3,4,5,6] in inhaled air. However, benzene is the compound that poses the greatest risk to workers’ health, due to its carcinogenic effects (Group I), according to the International Agency for Research on Cancer [2]. Filling station attendants, who perform various activities such as fueling, tank measurement, transfer of fuel from the truck to the storage tanks, administrative activities, and cleaning [7], are exposed both by inhalation and dermal routes. Although there is a benchmark for benzene concentration in gasoline, worker exposure is intense, as they have a workday of 9 h a day and 6 days a week.

In Brazil, in addition to the filling station attendants at gas stations, workers in convenience stores or even snack bars also often occupy the same space, increasing the number of workers exposed to fuel. Considering the context of gas stations in Brazil, occupational exposure may occur through direct contact (dermal or inhalation absorption) with gasoline, but also by breathing (inhalation of vapors) [2], depending on the service performed by the gas station workers.

Acute or chronic exposure to benzene may lead to a series of signs, symptoms, and complications which are characteristic of benzene poisoning [8], known in Brazil as benzenism, as an occupational disease. The most frequent changes include headache, somnolence, dizziness, tremors, mental confusion, asthenia, myalgia, and repetitive infection as the main signs and symptoms [9]. In addition, contact with epithelial tissue, eyes, and mucosa can cause irritation and tissue damage [9], allowing even greater dermal absorption of benzene and other VOCs present in fuels. Leukopenia, thrombocytopenia, eosinophilia, lymphocytopenia, monocytopenia, increased mean corpuscular volume (macrocytosis), and changes in neutrophils (neutropenia, basophilic stippling, neutrophil hyposegmentation) are the most frequent hematological changes [8,10]. In Brazil, suspected cases of hematological alterations related to benzene must be investigated using a hemogram with quantitative and qualitative analysis of the three-blood series [8].

Given the adverse health problems caused by benzene, this study aimed to evaluate the presence of signs and symptoms related to occupational exposure and whether occupational exposure to fuels is associated with the development of hematological changes in gas station workers in the city of Rio de Janeiro, Brazil.

## 2. Materials and Methods

### 2.1. Design and Study Population

This is a cross-sectional epidemiological study. Workers from gas stations in the city of Rio de Janeiro, Brazil, were recruited and interviewed after signing the informed consent form for the study. All gas station workers (GSWs) were invited to participate: both gas station attendants and managers—who incur inhalation and dermal exposure to the fuels and supply vehicles, receive and/or collect tank truck samples, and perform tank measurements underground of the station; and convenience store workers—who incur exposure to the fuels by inhalation (employees working in the administrative sector, cleaning, or in convenience stores).

To characterize the hematological changes from BTX, the exposed workers who presented hematological alterations performed two more collections within 15-day intervals (serial collection workers, SCWs). Workers with no occupational exposure to fuels were also selected to form the comparison group who were not occupationally exposed—office workers (OWs)—in the same region as the gas station.

Thirty-eight gas stations were visited, but after refusals and withdrawal of participation, 21 gas stations were included, 9 of which were located in the Central region of Rio de Janeiro and 12 in the South Zone. These regions were chosen because gas stations are concentrated in the same geographical region, thus, environmental and social issues were expected to be more homogenous. In addition, short distances between sites facilitated interviews and sample collection. Out of these, 324 GSWs participated in the study and 20 workers were included in the SCWs, regardless of their function.

For the OW group, 218 workers enrolled from the National Cancer Institute (INCA) and the Federal University of the State of Rio de Janeiro (UNIRIO), with no occupational exposure to benzene. In all cases, workers of both sexes participated in this study, who were over 18 years old and had at least 6 months of employment or contract.

This work was submitted and approved by the Human Research Ethics Committee of INCA (CEP/INCA) under No. 121/09. All workers invited to participate in the study signed an informed consent form prior to any intervention. The results obtained were delivered to individual workers, and if any changes in health conditions were perceived, the workers were referred for medical follow-up.

### 2.2. Collection of Data and Blood Samples

#### 2.2.1. Application of the Questionnaire

The study participants answered a standardized questionnaire applied through an interview with a trained professional. Information regarding socioeconomic aspects and chemical agent exposure during the workday were collected. A clinical questionnaire on current and prior history of complaints, signs, and symptoms was applied. Occupational habits were also evaluated.

The following variables were evaluated: sex; age; smoking; alcohol consumption; processed food consumption; working time; headache; weight loss; drowsiness; dizziness; tremor; loss of muscular strength; weakness; tingling; involuntary movements; petechial; bruising; epistaxis, and night sweats. The participants answered questions about common practices of the GSW: (1) clothes wet with fuel; (2) siphoning fuel with a hose; (3) trusting the automatic nozzle; (4) sniffing the fuel tank cap; (5) bringing their face closer to the fuel tank; (6) using a cloth.

#### 2.2.2. Biological Sample

The peripheral blood was collected by a professional through venipuncture of the forearm in a vacuum collection tube (EDTA anticoagulant) and kept refrigerated at approximately 8 °C in a Styrofoam box. The samples were then sent to the Laboratory of Clinical Pathology of the INCA for hematological analyses. This analysis considered gas station workers (GSWs) (*n* = 304), serial collection workers (SCWs) (*n* = 20), and office workers (OWs) (*n* = 218).

To confirm workers’ exposure to BTX, urine samples were collected at the end of the working day from all 8 h shifts (departure at 6 am, 2 pm, and 10 pm). Samples were collected once for each worker and the samples were analyzed separately. After collection, the urine samples were placed in a Styrofoam box containing recyclable ice, kept cold, and transported to the laboratory. Samples from the OWs (not occupationally exposed to benzene; comparison group) were collected at the end of the working day and manipulated in a similar way as samples from the exposed group (GSWs). For these analyses, GSWs (*n* = 255), SCWs (*n* = 20), and OWs (*n* = 100) were considered because some participants had creatinine urinary levels higher than the limit value.

### 2.3. Hematological Analysis

The blood samples were evaluated in an automated system (Sysmex XE-2100 system, São Paulo, Brazil). The total cell count parameter was selected, in which quantification of white blood cells, differential count of leukocytes (total leukocytes, neutrophils, lymphocytes) as well as erythrocytes, hemoglobin, hematocrit, and platelets were performed for all study participants.

### 2.4. Biomarkers of Exposure to Benzene

The urinary levels of trans, trans-muconic acid (tt-MA-metabolite of benzene) were evaluated by high efficiency liquid chromatography, a method initially proposed by Ducos et al. [11], modified by Paula et al. [12] using a UV detector (HPLC-UV) equipped with a 5 μm Luna Phenomenex^®^ chromatography column (250 × 4.6 mm) (Phenomenex^®^, Torrance, CA, USA). The standard was obtained from Sigma-Aldrich^®^ (Brondby, Denmark), and results were normalized using urinary creatinine values. Samples were cleaned by solid phase extraction (SPE) in Applied Separations^®^ cartridges (Applied separations ^®^ Allentown, PA, USA), N+ Quaternary Amino (SAX), 500 mg, 3 mL. The SPE conditioning was as follows: 3 mL of methanol, 3 mL of ultrapure water. One milliliter of urine was added, followed by a prewashing of 3 mL 1% acetic acid. Finally, elution occurred in 10% acetic acid (pH 2.7). Subsequently, an aliquot of 20 μL was injected into the HPLC by manual injection and the chromatographic conditions described by Geraldino et al. [13].

### 2.5. Biomarkers of Exposure to Toluene and Xylene

The urinary levels of hippuric acid (HA-metabolite of toluene) and Methyl Hippuric Acid (MHA-metabolite of xylene) were evaluated by high-performance liquid chromatography (HPLC) using an ultraviolet detector (HPLC-UV) equipped with a Lichrosorb RP18 chromatography column (244  ×  4 mm) with Merck 5 mm particles^®^ (Merck^®^, Darmstadt, Germany) [14]. The standard was obtained from Sigma-Aldrich^®^ (Brondby, Denmark), and results were normalized using the urinary creatinine levels. One milliliter of sample was added to 1 mL of methanol and then centrifuged for 7 min at 3000 rpm, with subsequent manual injection of 20 µL into the HPLC-UV.

### 2.6. Analytical Validation Parameters of Biomarkers

The validation parameters followed the recommendations by ANVISA [15]. Precision was obtained by calculating the coefficient of variation (relative standard deviation (RSD), %) of the peak area ratios of the seven injected replicates. The calibration curve was calculated from the concentrations obtained by the chromatograms, where a minimum of seven concentrations were used. For the validation stage, the limit of detection, limit of quantification and the linearity of the curve were obtained. Stability, recovery and matrix effect studies were also conducted.

#### 2.6.1. tt-MA

For the construction of the calibration curve, a linear range of tt-MA ranging from 0.01 to 100 mg/L was used. The RSD was less than 3%. The limit of detection and quantification were, respectively, 0.00075 and 0.0013 mg/L. The linearity coefficient was r = 0.996, within the acceptable range. The cartridge recovery study proved adequate (98–102%). The matrix effect and stability study were adequate.

#### 2.6.2. HA

The RSD estimate was less than 2% and a linear range of HA concentration from 0.01 to 10 g/L was observed. Seven points were used to construct the calibration curve, whose linearity coefficient had an acceptable performance (r = 0.998). The limit of detection and limit of quantification were, respectively, 0.0075 and 0.001 mg/L.

#### 2.6.3. MHA

The linearity coefficient remained within the acceptable limit range (r = 0.999). The RSD was less than 2% and the linear range of MHA concentrations was 0.001 to 10 g/L. Seven points were used in the construction of the calibration curve. The limits of detection and the limit of quantification were, respectively, 0.0005 and 0.005 mg/L.

### 2.7. Statistical Analysis

A descriptive analysis of the characteristics of the study population was carried out using a frequency distribution of the categorical variables, comparing the two groups of workers (GSWs and SCWs) with each other and to the OWs by the Chi-square test method. For the continuous variables, the measures of central trend and dispersion were calculated. The normality of the continuous variables was evaluated using the Kolmogorov–Smirnov test. As the variables presented a non-normal distribution, they were analyzed with the nonparametric tests of comparison of the Mann–Whitney test. For the categorical variables, the Chi-square or Fisher test was employed. To confirm the relationship between variables, linear regression analysis was used. Statistical analyses were performed using the program Statistical Package for Social Sciences (SPSS, Chicago, IL, USA) for Windows, version 17.0.

## 3. Results

This study involved the participation of 324 GSWs and 218 OWs. Analysis of biomarkers of exposure to BTX was performed on urine samples from 258 GSWs, 17 SCWs, and 100 OWs because some workers had high limit values of creatinine and were excluded. The analysis of urinary tt-MA revealed that the GSWs and SCWs had exposure to benzene at work (0.28 mg/g and 0.23 mg/g of urinary creatinine) when compared with the OWs (0.13 mg/g of urinary creatinine). Similar results were found for toluene and xylene that presented a higher mean of HA and MHA in the GSWs and SCWs than in the OWs, as can be seen in Table 1.

Exposure biomarkers can be influenced by confounding variables. For this reason, a linear regression analysis was performed considering smoking, alcohol, and processed food consumption as confounding factors. For tt-MA, the adjusted model was not affected by smoking (former smoker, *p*-value 0.965; smoker, *p*-value 0.466), alcohol consumption (*p*-value 0.638), and processed food consumption (three to four times a week, *p*-value 0.727; five times or more, *p*-value 0.318). For HA, the adjusted model was not affected by smoking (former smoker, *p*-value 0.803; smoker, *p*-value 0.669), alcohol consumption (*p*-value 0.189), and industrialized food consumption (three to four times a week, *p*-value 0.963; five times or more, *p*-value 0.260). For MHA, the adjusted model was not affected by smoking (former smoker, *p*-value 0.254; smoker, *p*-value 0.834), alcohol consumption (*p*-value 0.222), and industrialized food consumption (three to four times a week, *p*-value 0.098; five times or more, *p*-value 0.653). Therefore, no influences of confounding variables were found in the concentration of the analyzed biomarkers.

Table 2 presents the sociodemographic characteristics of the participants. Men comprised the majority of GSWs (69.4%) and SCWs (65%). The OWs were 58.3% women. The mean age of the GSWs was 35 years old (range = 20–70 years old), similar to the OWs (39 years, range = 19–68 years old). The mean age of SCWs was 27 years old (range 20–60 years old). For the life habits assessed, smoking was more frequent among the SCWs (35%), followed by GSWs (12.5%), and OWs (7.8%). Alcohol consumption was similar for the GSW and OW groups. Processed food consumption was similar for all groups.

The main occupations reported by GSWs were filling station attendant (47.2%), convenience store employee (25.3%), and manager/employer in charge (15.1%). Their most commonly reported length of work was up to ten years for 81.1% of GSWs and 70% of SCWs (Table 2). Some concerning occupational habits reported by filling station attendants and managers/employers were using cloths and flannels when fueling (73.5%), bringing their face closer to the fuel tank when filling up (40.9%), using clothes wet with fuel during their working time (29.3%), sniffing the fuel tank cap (25.6%), and sucking the fuel with their mouth using a hose (19.1%) (Figure 1).

The most described signs and symptoms of the GSWs, including convenience store employees, were drowsiness (45.1%), headache (38.3%), dizziness (27.5%), tingling (25.4%), involuntary movement (25%), and weakness (21.9%) as can be seen in Figure 2.

The hematological results showed that GSWs had significantly higher levels of erythrocyte, hemoglobin, hematocrit, and lymphocyte, and lower levels of neutrophil, than OWs (Table 3). The observed median results were in accordance with the reference value, despite being primarily located at the upper limit, except for erythrocyte, which had a median value higher (4.9) than the reference limit (3.8–4.8). When SCWs were evaluated, their leukocyte result was higher than both the OWs and GSWs, while their lymphocyte number was lower than that observed for the OW group (Table 3).

To analyze benzene poisoning, the important parameters to monitor in the hemogram include leukocyte and lymphocyte counts. Those analyses were performed three times with 15-day intervals. Of the 20 volunteers who presented hematological alterations, 5 remained with the same alterations with an interval of 45 days between collections. These exposed workers presented leukocytosis in the three collections and the number of lymphocytes was very close to the lower limit (Table 4).

## 4. Discussion

This study indicated that GSWs with different jobs in the same gas station environment, such as the filling station attendants, manager/employer, convenience store workers/employees, cleaners, and the administrative sector, are exposed to a high concentration range of BTX vapors in the workplace. There is concern worldwide to reduce, as much as possible, the exposure of these workers to benzene. The National Institute for Occupational Safety and Health (NIOSH) [16] established the value of 0.1 ppm (0.32 mg/m^3^), while the ACGIH established the value of 0.5 ppm (1.6 mg/m^3^) for the TLV-TWA, which corresponds to the occupational exposure limit value, which takes into account a weighted average over an 8 h working day and a 40 h working week [17]. In Brazil, occupational exposure to benzene is regulated through Regulatory Standard No. 15 [18], which determines “technological reference values” with maximum levels of 2.5 ppm for steel industries and 1.0 ppm (3.19 mg/m^3^) for chemical and petrochemical sectors, such as gas stations.

Cruz et al. [19] determined the concentrations of BTEX compounds present in the air at ten gas stations in the state of Bahia, Brazil. Toluene, ethylbenzene and xylenes were present in concentrations lower than the limits recommended by NIOSH and NR-15 of the Brazilian Ministry of Labor. However, benzene concentrations at 3 stations (out of 10 evaluated) were above the exposure limit recommended by NIOSH. For benzene, although the average of the evaluations was 0.21 mg/m^3^, a variation between 0.04 mg/m^3^ and 0.43 mg/m^3^ was observed. Thus, 33% of the stations had an environmental concentration of benzene greater than 1.0 ppm [5].

In this study, the environmental monitoring of exposure to BTX was not performed, but exposure biomarkers present in the urine of workers were analyzed. According to Brazilian legislation, urinary tt-MA is indicated as a biomarker of benzene exposure. The correlation between tt-MA and benzene is detected at environmental levels of the solvent below 1.0 ppm; this fact allows its use as a biomarker of exposure in gas station workers, since the maximum concentration allowed in these environments is 1.0 ppm. Scherer et al. (1998) observed a linear correlation between urinary excretion levels of tt-MA and occupational exposure to benzene in workers exposed to concentrations above 0.1 ppm [20]. As a result, urinary tt-MA has been used in the occupational assessment of benzene. Based on these, the tt-MA was analyzed as a biomarker of exposure to benzene, HA for toluene, and MHA for xylene. The GSWs had statistically higher mean values of tt-MA, HA, and MHA than those presented by the OWs, as expected.

Some habits may interfere with urinary tt-MA, HA, and MHA levels such as smoking, consumption of processed foods containing sorbic acid (or any of its salts), and alcoholic beverages [21]. For this reason, multiple linear regression analyses were performed to verify which variables could be interfering with the results of the presented biomarkers. Our analyses did not show that such variables influenced the levels of the evaluated biomarkers. Considering that the difference in biomarkers found between the GSWs and OWs was not influenced by workers’ habits, the GSWs were then considered exposed and the OWs considered unexposed.

Some occupational habits that were identified during the interview could increase exposure to benzene present in gasoline through its contact with skin and mucous membranes. During analysis of the frequency of the main occupational habits or procedures that could enhance exposure, 29.3% of workers reported that they had worked in uniforms wet with fuel. This often occurs after a fuel spill or improper handling of fuel pumps. Being in contact with a piece of clothing soaked with fuel increases contamination through dermal absorption and consequently increases exposure [10,22]. Research in the literature has demonstrated that the frequent change of uniforms of workers in a coke oven reduces the absorption of benzene by the body [23].

In Brazil, filling vehicles can only be conducted with the automatic filling spout. They have a lock that, among other functions, prevents spillage/leakage of fuel during fueling. It was observed that 53% of the workers trust the automatic filling nozzle of the fuel pump, but the remaining 47% do not trust it. When fuel spills occur during refueling, filling station attendants usually use a cloth to clean the vehicle. Many of them keep this cloth close to their body, on their shoulders or waist so that it can be used promptly in case a spill occurs during another filling [22]. It is known that the absorption of benzene by the skin differs according to the body part. The back/forearm region has medium absorption of benzene. The scrotum/waist region and forehead/ head region have high absorption [24]. The cloth contaminated with fuel is usually held close to these areas and can increase the exposure of these workers.

Brazilian legislation set 2033 as a deadline for gas stations to install vapor recovery systems at their pumps to reduce workers’ exposure to gasoline components that evaporate during filling [25]. Moreover, Annex 2, Regulatory Standard No. 9 [26] states that all workers should have another uniform in case of an accident (including soaking with fuel); in addition, the employer should wash the uniforms to minimize exposure to workers and their families. However, this is not routinely undertaken, especially laundering the uniform, bringing more risk to the workers [27].

Other occupational habits that increase contact with fuels were also reported by the filling station attendants. The habit of siphoning fuel with the mouth occurs when the attendant fills the vehicle with the wrong fuel and the worker uses this mechanism to remove the wrong fuel from the tank. It is known that siphoning through the oral mucosa is increased and, therefore, the filling station attendants who reported performing this activity may increase their exposure [10].

Gasoline has a sweet smell characteristic of benzene [9], and this smell becomes attractive to some workers who acquire the habit of sniffing the fuel cap. This habit is worrisome because it is known that exposure through inhalation is important and sniffing the cap enhances this exposure [10]. Another important habit is putting their ear close to the fuel tank to check the supply. BTX can also cause hearing changes and bringing ears closer to the fuel tank can increase exposure [10].

The main occupation reported in the exposed groups was filling station attendant. This result was similar to that reported by Amaral et al. [28], who observed filling station attendant as the main occupation. According to the Brazilian Classification of Occupations [29], filling station attendants have the function of attending gas stations, being able to perform, among other activities, the following: fuel vehicles; check level and perform fluid exchange of vehicles; and check quantity, list, order and receive goods, in addition to examining their condition (general aspects and validity). These activities contribute to the emission of vapors present in fuels (BTX) in the environment. However, not only filling station attendants who carry out these activities are exposed. Other workers working at gas stations are also exposed since the BTX vapors can be inhaled because they are in the same area of work. Therefore, employees of the cleaning service, administrative sector, and convenience stores (snack bars) are expected to be exposed to BTX vapors too.

According to Fundacentro (Jorge Duprat Figueiredo Foundation for Occupational Safety and Medicine, Brazil), benzene poisoning is characterized by the presence of signs, symptoms, and complications after acute or chronic exposure to benzene, and chronic exposure can promote diverse signs and symptoms, with the possibility of preferential complications in the hematopoietic system [10]. The OWs’ signs and symptoms were not analyzed in the present study because those symptoms may be due to other factors impacting office jobs, such as extensive use of computers [30].

The main signs and symptoms of the GSWs were analyzed, and the literature indicates that most effects related to acute benzene intoxication are mucosal irritation; central nervous system effects, such as drowsiness, dizziness, excitation, headache; tremors; loss of consciousness; and death [6,8,23]. In this study, the most reported symptoms were drowsiness (45.1%), headache (38.3%), dizziness (27.5%), tingling (25.4%), involuntary movement (25%), and weakness (21.9%). Most of the signs and symptoms presented are related to occupational exposure to benzene and may be associated with benzene poisoning, especially when hematological changes are present [10].

Since there is no established effect biomarker for benzene in Brazil, the present article evaluated the levels of exposure of workers to benzene through tt-MA, the presence of signs and symptoms related to occupational exposure, and whether occupational exposure to fuels with BTX is associated with the development of hematological changes in gas station workers.

Major hematological changes in benzene poisoning include persistent leukocytosis; mean corpuscular enlargement (macrocytosis); eosinophilia; decrease in the absolute number of lymphocytes (lymphopenia or lymphocytopenia); neutrophilic changes: basophilic stippling, hyposegmentation of neutrophils (pseudo-Pelger); presence of macroplatelets; and leukopenia with association of other cytopenias (thrombocytopenia) [10]. In this study, erythrocytes, hemoglobin, hematocrit, total leukocytes, neutrophils, lymphocytes, and platelets were analyzed.

The hematological analysis revealed that of the 324 GSWs whose blood was collected, 304 maintained their results within the reference limits and, for this reason, did not participate in the serial collection. The median values presented by the GSWs were similar to those presented by the OWs; nevertheless, a statistical difference occurred between the groups for erythrocytes, hemoglobin, hematocrit, neutrophils, and lymphocytes.

Data from the literature revealed several types of changes in the hematopoietic system in workers with benzene exposure, either with an increase or decrease in the parameters evaluated. In contrast with our study, many studies about gas station workers showed a decrease in the number of erythrocytes and hemoglobin among those exposed [31,32,33,34]. However, Ahmadi et al. [35], similar to our study, found enhancement of erythrocyte, hemoglobin, and hematocrit in filling station attendants. They believed that workers had greater levels of oxidative stress because antioxidant power was significantly lower in the exposed group than the control group [35].

Of the 20 workers who performed the serial collection, 5 workers remained with an increased number of leukocytes and a low number of lymphocytes, close to the lower limit of the reference value. A previous study observed that some GSWs, regardless of their occupation, were more likely to present DNA damage and alterations in cells of the immune system than workers belonging to the comparison group without occupational exposure to BTX [36].

According to the Health Surveillance Standard for workers exposed to benzene, they must be workers who present hematological alterations in serial blood counts, without other clinical findings that justify them, among others: persistent leukocytosis and lymphopenia. In situations where the alterations persist in this minimum period of 45 days, the suspected case for benzene poisoning is considered [37]. Chronic exposure to benzene may lead to changes in bone marrow, blood, immune system, and various types of cancer [10]. Thus, the alterations observed in this study could be an initial change that can contribute to leukemogenic risk.

There is little investment in safety measures and training of gas station workers in Brazil. In addition, other factors, such as high temperatures throughout most of the year, and tampering with gasoline with higher concentrations of benzene and other solvents than permitted by law, increase the exposure level and the risk of disease. This should be aggregated with information about the presence of signs and symptoms related to occupational exposure and whether occupational exposure to fuels with BTX is associated with the development of hematological changes in gas station workers in the city of Rio de Janeiro, Brazil, in order to contribute to health surveillance strategies of benzene-exposed workers in the country.

The observed results reveal initial changes in parameters routinely used to evaluate the health conditions of the population, when different groups are compared. This can help to reinforce adoption of practical measures in worker health surveillance, such as recognizing clinical changes even in the normal laboratory results, considering the care and monitoring of the health of GSWs, as well as other groups of workers who share the same spaces as the service stations.

## 5. Conclusions

This study showed that occupational exposure of gas station workers by inhalation and/or dermal route increased the frequency of signs and symptoms related to benzene intoxication such as somnolence, headache, dizziness, tingling, involuntary movements, and weakness. Hematologic changes such as leukocytosis and lymphopenia were also present in the serial collection group. Although concentrations of benzene and other solvents in gas stations are considered low, continuous exposure may lead to chronic effects for workers who are in contact with these compounds during their workday, causing the greater risk of developing hematological damage by these workers and possibly for nearby residents. The results of this article may contribute to technical and scientific support to raise discussions around the theme, and contribute to the establishment of stricter standards for the reduction/elimination of human exposure to carcinogens, such as benzene, so that effective surveillance and control measures in work environments can be taken routinely and consistently. We hope that these results can contribute to the prevention and surveillance of cancer and other work-related diseases, since the monitoring of workers exposed to agents with high human toxicity are pillars of the surveillance of populations exposed to carcinogens. Therefore, our findings corroborate the importance of periodically monitoring the health of gas station workers, including convenience store employees.

## Figures and Tables

**Figure 1 ijerph-20-05896-f001:**
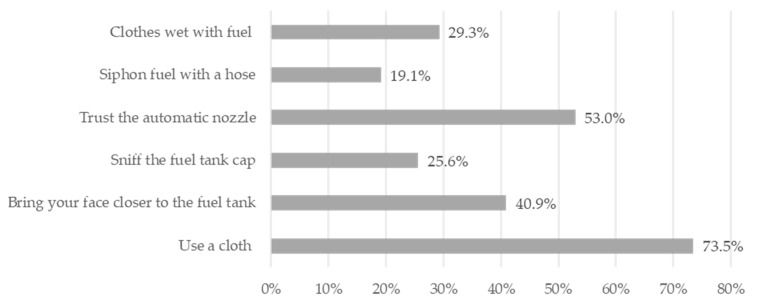
Occupational habits of filling station attendants and managers/employers (*n* = 215). City of Rio de Janeiro, RJ-Brazil, 2014–2016.

**Figure 2 ijerph-20-05896-f002:**
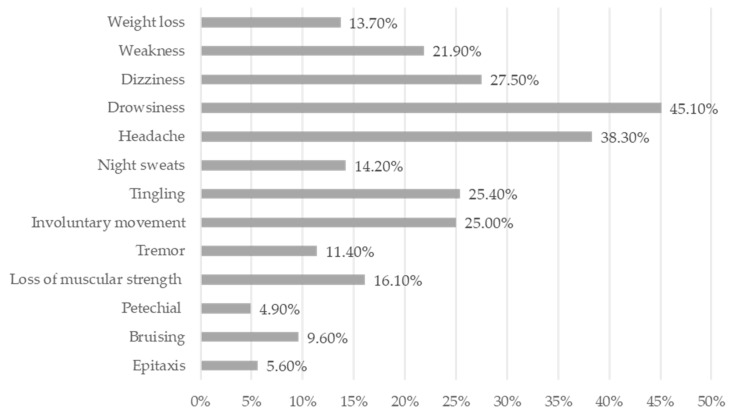
Signs and symptoms related to benzene poisoning in gas station workers (*n* = 324). City of Rio de Janeiro, RJ-Brazil, 2014–2016.

**Table 1 ijerph-20-05896-t001:** Analysis of tt-MA, HA, and MHA of the studied population (*n* = 375). City of Rio de Janeiro, RJ-Brazil, 2014–2016.

	Exposed		Not Exposed	*p*-Value ³
	Gas Station Workers (GSW) *n* = 258	Serial Collection Workers (SCW) *n* = 17	Office Workers (OW) *n* = 100	
Exposure Biomarker				
tt-MA ^1^	0.28 (0.62)	0.23 (0.23)	0.13 (0.22)	0.001
HA ^2^	0.50 (0.29)	0.40 (0.31)	0.08 (0.19)	<0.001
MHA ^2^	1.57 (1.44)	1.73 (1.42)	0.02 (0.05)	<0.001

Values presented in mean (standard deviation). ^1^ Values presented: mg/g urinary creatinine; ^2^ Values presented: g/g urinary creatinine. ^3^ Mann–Whitney test.

**Table 2 ijerph-20-05896-t002:** Sociodemographic characteristics and life habits of the studied population (*n* = 542). City of Rio de Janeiro, RJ-Brazil, 2014–2016.

	Exposed		Not Exposed
	Gas Station Workers (*n* = 304) ¹ *n* (%)	Serial Collection Workers (*n* = 20) ² *n* (%)	Office Workers (*n* = 218) ³ *n* (%)
Sex			
Male	211 (69.4)	13 (65.0)	91 (41.7)
Female	93 (30.6)	7 (35.0)	127 (58.3)
Age ^4^	35 (20–70)	27 (20–60)	39 (19–68)
Smoking			
Non-smoker	218 (71.7)	10 (50.0)	169 (77.5)
Former smoker	48 (15.8)	3 (15.0)	32 (14.7)
Smoker	38 (12.5)	7 (35.0)	17 (7.8)
Alcohol consumption			
No	114 (37.5)	4 (20.0)	76 (34.9)
Yes	190 (62.5)	16 (80.0)	142 (65.1)
Processed food consumption			
Once or twice a week	113 (39.6)	8 (42.1)	75 (39.5)
Three or four times	48 (16.8)	2 (10.5)	35 (18.4)
Five times or more	124 (43.5)	8 (42.1)	80 (42.1)
Working time			
≤10 years	245 (81.1)	14 (70.0)	133 (73.5)
10 to 20 years	34 (11.3)	3 (15.0)	32 (17.7)
>20 years	23 (7.6)	3 (15.0)	16 (8.8)

Values expressed as absolute number and frequency; ^1^ gas station workers; ^2^ serial collection workers; ^3^ office workers; ^4^ values presented in median (min–max).

**Table 3 ijerph-20-05896-t003:** Hematological and biochemical parameters in the study population (*n* = 542). City of Rio de Janeiro, RJ-Brazil, 2014–2016.

Hematology	Reference Value	Exposed		Not Exposed
		Gas Station Workers ^1^ *n* = 304	Serial Collection Workers ^2^ *n* = 20	Office Workers ^3^ *n* = 218
Erythrocyte (×10^3^/µL)	3.8–4.8	4.9 (3.5–6.3) ¹	4.7 (4.2–5.3)	4.7 (3.71–6.0)
Hemoglobin (g/dL)	12.0–15.0	14.5 (9.3–17.6) ¹	14.2 (11–16.2)	13.8 (10.0–17.5)
Hematocrit (%)	36–46	43.0 (30.2–52.9) ¹	42.2 (33.8–48.5)	41.6 (33.7–50.5)
Leukocyte (U/µL)	4000–10000	7270 (3070–14130)	10,705 (4300–13,600) ^2,3^	7005 (2780–20,400)
Neutrophil (%)	40–75	56.3 (22.3–78.1) ¹	58.3 (41.3–74.9)	58.4 (28.0–85.0)
Lymphocyte (%)	20–45	32.8 (11.9–58.9) ¹	27.7 (18.2–47.3) ³	31.7 (9.1–57.9)
Platelets (×10^3^/µL)	163–343	253 (122–573)	279 (176–540)	245 (93–534)

Values expressed as median (min–max). ^1^ GSWs different from OWs; ^2^ SCWs different from OWs; ^3^ SCWs different from GSWs.

**Table 4 ijerph-20-05896-t004:** Individual results of the three serial collections carried out on workers from gas stations in Rio de Janeiro, RJ-Brazil, 2014–2016.

	Leukocyte (U/µL)			Lymphocyte (%)		
¹ COD	First Sample	Second Sample	Third Sample	First Sample	Second Sample	Third Sample
E004	11,390	11,210	11,730	22.7	23.1	21.8
E040	10,970	12,160	11,700	28.2	27.4	25.0
E216	11,450	11,410	11,500	25.0	23.8	27.7
E256	11,020	10,930	10,320	23.5	28.9	24.0
E280	10,300	11,390	11,470	32.3	22.5	23.3

^1^ Workers’ coding. *n* = 5 participants.

## Data Availability

Data are not publicly available due to ethical criteria restrictions. So, the data presented in this study are only available in form of statistical data.

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
