# Peer review of "Hematological Changes in Gas Station Workers"

_ijerph, 2023, doi:10.3390/ijerph20105896_

Round 1

Reviewer 1 Report (New Reviewer)

Author Response

REPLIES TO REVIEWERS

This study investigates hematological and biochemical changes in gas-stations workers and in workers without occupational exposure in order to evaluate the presence of signs and symptoms related to gasoline occupational exposure. The study is quite interesting, the quality of analysis and the idea are good, but a major revision is needed. In our opinion, hematological and biochemical output should be better related to occupational exposure. Additionally, some parameters related to method validation (e.g. LODs, LOQs, RSD%, linear range, etc.) should be also added as well.

  • Material and methods: For biomarkers of exposure (t,t-MA, hippuric acid, Methyl Hippuric Acid) please add LODs, LOQs, RSD%, and linear range.
  • Results-Discussion: The authors describe some worker behaviours that may increase the level of exposure to benzene through routes other than inhalation (e.g. dermal or oral). However, in our opinion, the absence of benzene concentrations in the environment does not allow for complete characterization of the exposure, at least through inhalation. In addition to considerations made about other routes of uptake, we suggest comparing the values found for t,t-MA with other values described in the literature and related to gas stations or gasoline exposure, when environmental values are also reported (e.g. in an additional Table). In this way, you might have some idea of the benzene concentrations in the environment that can determine the levels of biomarkers of exposure found.

Answer: The above inclusion requests were met and included in the manuscript.

Please revise the manuscript according to the referees' comments and upload the revised file by 24 March 2023.

Please use the version of your manuscript found at the above link for your revisions.

(I) Please check that all references are relevant to the contents of the manuscript.

Answer: Necessary articles were maintained and new articles were added based on reviewers' suggestions.

(II) Any revisions to the manuscript should be marked up using the “Track Changes” function if you are using MS Word/LaTeX, such that any changes can be easily viewed by the editors and reviewers.

Answer: Sorry for the text review not taking place with edit control. In any case, the altered passages were marked in red.

(III) Please provide a cover letter to explain, point by point, the details of the revisions to the manuscript and your responses to the referees’ comments.

Answer: Cover letter provided as requested.

(IV) If you found it impossible to address certain comments in the review reports, please include an explanation in your appeal.

(V) The revised version will be sent to the editors and reviewers.

This manuscript is a resubmission of an earlier submission. The following is a list of the peer review reports and author responses from that submission.

Round 1

Reviewer 1 Report

The study is very interesting and the paper is well writing. The methods section can be improved by addressing the following:

Design and study population

1. Clarify how the 38 filling stations were chosen as potential study sites- from which 21 were ultimately used for the study. 

2. Method used for the selection of study participants. were gas station workers and CSW selected by random selection from the 21 gas stations used as study sites or was convenience sampling used. Please also justify the selection method used.

3. Same as 2 for control/ comparison group. 

2.2.2. Please clarify why urine sample were collected in 9 participants for gas station workers (out of 324)  and 8 for office workers (out of 218).

Results

There are so many abbreviations used , which confuses the reader. the full name for ASF and ACS was not stated. 

Discussion

Paragraph 2: Most participants did not report signs and symptoms. Rephrase the statement "Occupational exposure to fuel vapors showed in our study that exposed workers had a high frequency of self reported headaches, slimming, dizziness, weakness and tingling. Instead "Occupational exposure to fuel vapors showed that a high proportion of CSW workers (48.2%) and FSW workers (33.2%) reported headaches. CSW workers reported dizziness (41.8%, weakness (31.8%) and Tingling (32.1%) compared to FSW workers (p<0.05).

The authors showed emphasize the implication in reduction /increase or normal ranges of hematological and biochemical indicators studies. At the moment, the results are just compared to similar or related studies, but the implications of what the findings indicate is not clearly discussed. 

Conclusion

The fist statement should be revised to reflect the results reported. 

Author Response

Thank you so much for your precious review!
We accept all your suggestions and believe that they were important to improve the article.

Reviewer 2 Report

The study investigates biochemical effects of BTX aromatic compounds in occupationally exposed persons. Exposures to benzene are low for occupationally exposed persons and only slightly higher than in the cotrol group. The statistical analyses of the biochemical effects are only univariate (without taking into account gender, age, alcohol and tobacco consumption). The effects are difficult to interpret and are not indicative of effects caused by BTX.

e.g. exposed have higher values for erythrocytes, haemoglobin, haematocrit and lymphocytes, although the opposite would be expected for haematological effects caused by benzene. In the case of liver enzymes, gamma GT showed the greatest difference between exposed persons and controls. Here it should be checked (multiple linear regression) whether this is not an effect that can be attributed to the alcohol.

A subgroup analysis with n = 11, 8 and 6 exposed persons only makes sense if the effects from the study have proven to be robust.

Author Response

(The authors gave the same response as above.)

Round 2

Reviewer 2 Report

The statistical analyses of the biochemical effects are only univariate (without taking into account gender, age, alcohol and tobacco consumption). 

Author Response

The statistical analyses of the biochemical effects are only univariate (without considering gender, age, alcohol and tobacco consumption).

Answer: Dear reviewer, for the last version of the article, we reformulated the presentation of the results. Thus, we decided to withdraw the biochemical analyzes, bearing in mind that after the bivariate analysis, the results were no longer statistically significant. Thus, in order to produce a document with greater organization, objectivity and applicability, we chose to remove the biochemical analyzes. We believe that the new version has better quality, even without presenting information on biochemical effects.